# Human RPA activates BLM's bidirectional DNA unwinding from a nick

**Zhenheng Qin[1,2,3†], Lulu Bi[1†], Xi-Miao Hou[4], Siqi Zhang[1,2,3], Xia Zhang[1], Ying Lu[5], Ming Li[3,5], Mauro Modesti[6], Xu-Guang Xi[7]\*, Bo Sun[1]\***

[1]School of Life Science and Technology, ShanghaiTech University, Shanghai, China; [2]Shanghai Institute of Biochemistry and Cell Biology, Chinese Academy of Sciences, Shanghai, China; [3]University of Chinese Academy of Sciences, Beijing, China; [4]College of Life Sciences, Northwest A&F University, Yangling, China; [5]Institute of Physics, Chinese Academy of Sciences, Beijing, China; [6]Cancer Research Center of Marseille, CNRS UMR7258, Inserm U1068, Institut Paoli-Calmettes, Aix-Marseille Université, Marseille, France; [7]The LBPA, Ecole Normale Supérieure Paris-Saclay, CNRS, Université Paris Saclay, Gif-sur-Yvette, France

**Abstract** BLM is a multifunctional helicase that plays critical roles in maintaining genome stability. It processes distinct DNA substrates, but not nicked DNA, during many steps in DNA replication and repair. However, how BLM prepares itself for diverse functions remains elusive. Here, using a combined single-molecule approach, we find that a high abundance of BLMs can indeed unidirectionally unwind dsDNA from a nick when an external destabilizing force is applied. Strikingly, human replication protein A (hRPA) not only ensures that limited quantities of BLMs processively unwind nicked dsDNA under a reduced force but also permits the translocation of BLMs on both intact and nicked ssDNAs, resulting in a bidirectional unwinding mode. This activation necessitates BLM targeting on the nick and the presence of free hRPAs in solution whereas direct interactions between them are dispensable. Our findings present novel DNA unwinding activities of BLM that potentially facilitate its function switching in DNA repair.

**\*For correspondence:**
xxi01@ens-cachan.fr (X-GX);
sunbo@shanghaitech.edu.cn (BS)

†These authors contributed equally to this work

**Competing interests:** The authors declare that no competing interests exist.

## Introduction

DNA helicases are ubiquitous motor proteins that couple the hydrolysis of nucleoside triphosphates (NTPs) to the unwinding of double-stranded DNA (dsDNA), providing the single-stranded DNA (ssDNA) required for many biological processes, including DNA replication, repair and recombination (*Singleton et al., 2007*). Among these DNA helicases, the RecQ family helicases have been highly conserved during evolution from prokaryotes to humans and play critical roles in genome maintenance and stability (*Bernstein et al., 2010*; *Croteau et al., 2014*). Defects in three of the five human RecQ members (BLM, WRN and RECQ4) give rise to distinct heritable human disease syndromes (Bloom's, Werner's and Rothmund-Thomson syndromes, respectively), characterized by genomic instability and an increased incidence of cancers (*Ellis et al., 1995*; *Kitao et al., 1999*; *Yu et al., 1996*). These genetic linkages underscore the importance of the RecQ family helicases in cellular homeostasis. Thus, an in-depth understanding of their unwinding mechanisms at the molecular level may inform potential therapeutic strategies.

Bloom syndrome protein (BLM) is one of the five RecQ family helicases that unwind DNA in a $3'-5'$ direction (*Croteau et al., 2014*; *Karow et al., 1997*). It is involved in many aspects of genome maintenance, including DNA end resection (*Kowalczykowski, 2015*), displacement-loop (D-loop) processing (*Bugreev et al., 2007*), rescuing stalled or collapsed replication forks (*Davies et al., 2007*), and resolution of Holliday junction (*Bizard and Hickson, 2014*). BLM's diverse functions are

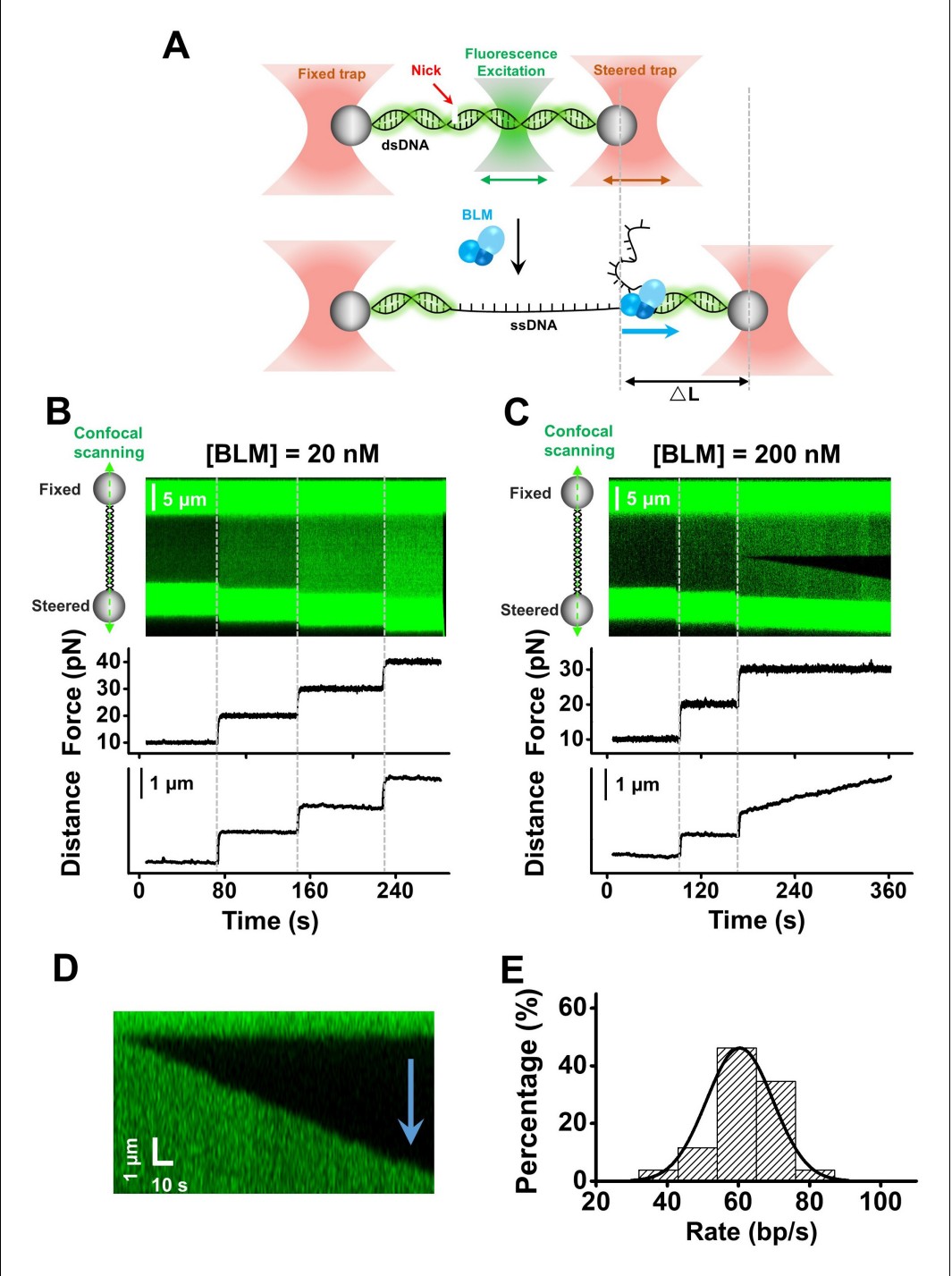

**Figure 1.** BLM unwinds dsDNA unidirectionally from a nick. (**A**) A Schematic of the experimental configuration. Biotinylated λ phage DNA was suspended by two streptavidin coated beads manipulated by two optical traps. Meanwhile, confocal lasers repeatedly scanned along the DNA template. DNA length was expected to increase under a constant force when dsDNA was converted to ssDNA by BLM. (**B, C**) A representative kymograph of a tethered λ DNA as well as its corresponding force and DNA length as a function of time in the presence of 20 nM (**B**) or 200 nM (**C**) BLM. (**D**) Kymograph of an individual unwinding event showing the unidirectional DNA unwinding. The arrow indicates the direction of the unwinding fork movement. (**E**) Distributions of the dsDNA unwinding rates in the presence of 200 nM BLM under 30 pN.

The online version of this article includes the following source data and figure supplement(s) for figure 1:

**Source data 1.** The dsDNA unwinding rates of 200 nM BLM under 30 pN.

*Figure 1 continued on next page*

*Figure 1 continued*

**Figure supplement 1.** DNA unwinding in the presence of 200 nM BLM.
**Figure supplement 2.** Control experiments verified BLM's unwinding activity.
**Figure supplement 3.** Elasticity parameters of ssDNA and DNA unwinding rate calculation.

realized by its remarkable unwinding capabilities of many unique DNA structures, such as G-quadruplex (*Chatterjee et al., 2014*). In particular, BLM has been verified to participate in many steps in repairing DNA double-strand breaks (DSBs) via homologous recombination (HR) (*Nimonkar et al., 2011*; *Nimonkar et al., 2008*; *Sturzenegger et al., 2014*; *Woglar and Villeneuve, 2018*). It has been demonstrated to be recruited to the DSB ends and facilitate resection of the 5' termini to generate 3' end protruding ssDNAs in the first step of HR (*Gravel et al., 2008*). BLM either unwinds dsDNA to provide ssDNA for exonuclease resection by EXO1 or DNA2 or directly stimulates their nucleolytic activities (*Nimonkar et al., 2008*; *Sturzenegger et al., 2014*). Paradoxically, even though BLM is distinctive in preferentially unwinding unique substrates (*Mohaghegh et al., 2001*), an in vitro ensemble study has demonstrated that this helicase alone is unable to initiate dsDNA unwinding from a nick which is a prevalent and essential DNA intermediate during HR (*Maizels and Davis, 2018*; *Mohaghegh et al., 2001*). These contradictory findings may be reconciled by BLM's varying unwinding activities, which are often regulated by other in vivo factors (*Croteau et al., 2014*).

It has been widely appreciated that the BLM helicase interacts and collaborates with many protein partners to aid in cellular responses to replication stress and DNA damage in vivo (*Croteau et al., 2014*). These interactions and collaborations dictate BLM's specialized functions in genome maintenance. A key partner of BLM is the single-stranded DNA-binding (SSB) protein, replication protein A (RPA), a heterotrimeric protein complex consisting of the RPA70, RPA32 and RPA14 subunits, which is also essential in DNA repair and recombination (*Fanning et al., 2006*; *Wold, 1997*). A recent in vivo study directly demonstrated that RPA co-localizes with BLM at the foci of DNA damage, suggesting that these two proteins coordinate and function together in DNA repair (*Woglar and Villeneuve, 2018*). In addition, biochemical, structural and single-molecule studies have revealed that RPA not only physically interacts with BLM but also stimulates its unwinding activity by increasing its processivity or improving its unwinding initiation (*Brosh et al., 2000*; *Doherty et al., 2005*; *Soniat et al., 2019*). The poor ability of a heterologous SSB protein to stimulate BLM's unwinding further suggests that this stimulation is specific (*Brosh et al., 2000*). These findings, coupled with the inability of BLM alone to unwind a nicked DNA in vitro, suggest that RPA might also promote BLM in unwinding nicked dsDNA to facilitate efficient end resection in HR. A thorough understanding of how BLM-mediated DNA unwinding is regulated by RPA would advance our knowledge in its diverse roles and functions in genome maintenance.

In this study, by combining optical tweezers with confocal fluorescence microscopy, we examined BLM unwinding activities under various conditions at the single-molecule level. We found that, in the absence of hRPA, BLM at high concentrations can unwind dsDNA from a nick unidirectionally with a requirement of an external destabilizing force on the DNA template. Surprisingly, the presence of hRPA permits BLM's unwinding in two opposite directions from a nick at low and zero forces. This stimulation demands free hRPAs in solution, suggesting that the coating of hRPA on the newly generated ssDNA is essential for enhancing BLM unwinding activity. These findings reveal a distinct SSB-enabled helicase unwinding mode that might facilitate the generation of the 3' ssDNA tail in HR. Additionally, this study demonstrates various DNA unwinding activities by BLM that are regulated by tension on DNA, the concentration of BLM, and the presence of RPA, which provide a molecular insight into how BLM realizes its diverse biological functions.

## Results

### Force assists BLM's unidirectional unwinding from a nick

We combined dual-optical traps with confocal microscopy to monitor unwinding of a single DNA molecule by wild type (WT) *Gallus gallus* BLM (gBLM). gBLM (hereafter referred to as BLM) has two similar RPA binding domains to the human BLM (hBLM) and the sequence of its helicase-core domain (core-gBLM) has an 80% identity with the one from hBLM. *Figure 1A* shows a schematic of

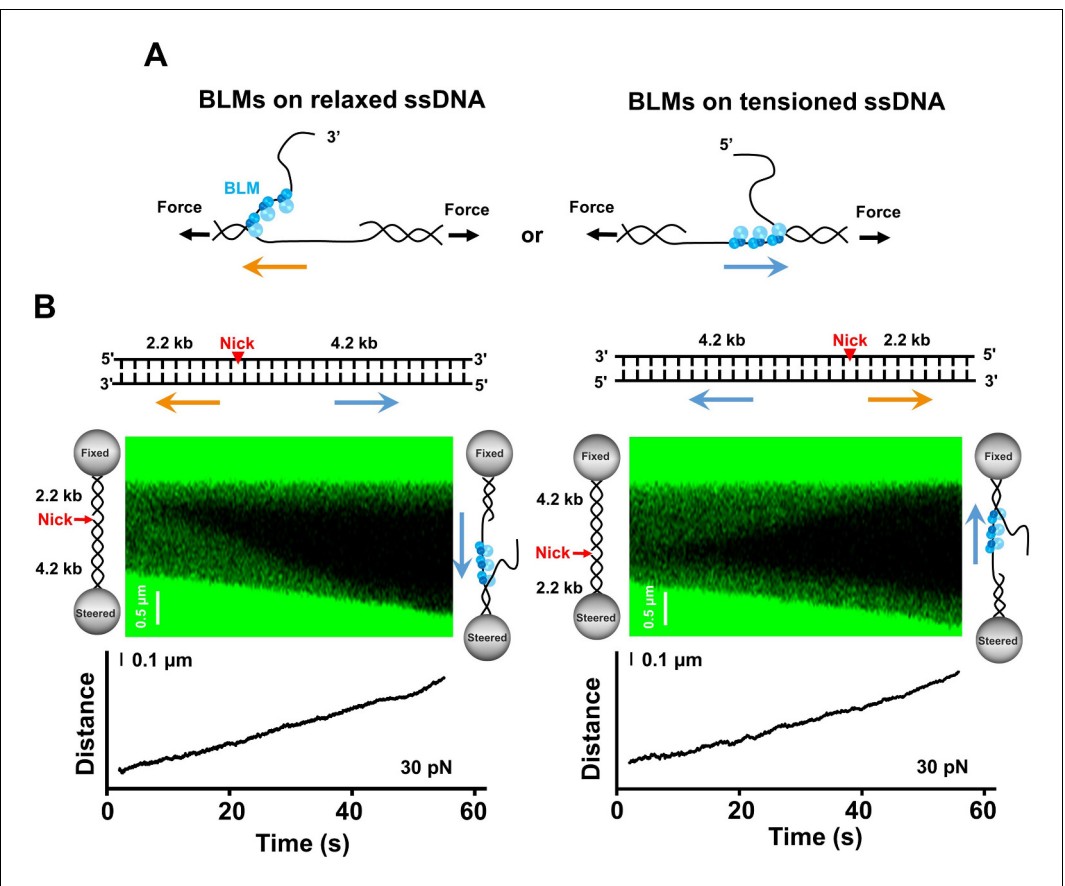

**Figure 2.** BLM unwinds DNA template containing a single nick. (**A**) Two models of BLM unwinding dsDNA from a nick. BLM initiates unidirectional dsDNA unwinding from a nick by translocating either on the tensioned ssDNA or on the relaxed ssDNA. The unwinding polarities are opposite in these two scenarios. (**B**) The 6.4 kbp dsDNA template unwinding in the presence of 200 nM BLM. The 6.4 kbp dsDNA contains a nick located at 2.2 kbp and 4.2 kbp from the 5' and 3' termini, respectively. Representative kymographs and DNA length signals showed the 4.2 kbp dsDNA segments were always unwound under 30 pN. The arrows indicate the directions of the unwinding fork movement.

The online version of this article includes the following figure supplement(s) for figure 2:

**Figure supplement 1.** Construction of the 6.4 kbp DNA template.

our experimental configuration in which a λ phage DNA molecule is suspended between one fixed trap and one steered trap via two streptavidin-coated microspheres, while a confocal laser repeatedly scanned along the plane of the DNA template (*Figure 1A*). A high-frequency feedback system on the steered trap was employed to ensure the force on the DNA template remained constant, while BLM unwound the DNA template. The difference in extension between ssDNA and dsDNA under tension would result in a change in DNA length once the BLM helicase started unwinding (*Bustamante et al., 2003*), thus allowing for monitoring of its unwinding activity. In addition, the fluorescent binding agent Sytox was used as a dsDNA probe, and its immediate dissociation led by converting dsDNA to ssDNA by BLM served as an additional signal for dsDNA unwinding. This combined single-molecule technique allowed us not only to record the change of DNA length induced by BLM-catalyzed dsDNA unwinding but also to directly visualize individual unwinding events and DNA intermediates along the DNA template.

We started the unwinding experiment at a force of 10 pN for ~1 min and increased it by 10 pN each time until dsDNA unwinding was detected. In the presence of 20 nM BLM, apparent dsDNA unwinding was not observed over a broad force range from 10 pN to 40 pN (*Figure 1B*). In contrast, at a high BLM concentration of 200 nM, one or few unwinding events on a single DNA molecule were recorded once the force was increased to a range of 10–40 pN in all examined traces (n = 30)

(*Figure 1—figure supplement 1*). *Figure 1C* shows a kymograph of a DNA molecule and the force and DNA length as a function of time in the presence of 200 nM BLM: DNA length and fluorescence signals did not change under 10 or 20 pN; however, the DNA length displayed a continuous increase while a dark region in the middle of the fluorescent dsDNA appeared and subsequently expanded just after the force was increased to 30 pN. Omitting ATP did not result in changes in either fluorescence or DNA length signal in this experimental condition, confirming that our observations were a result of BLM-catalyzed dsDNA unwinding (*Figure 1—figure supplement 2A*). Control experiments without Sytox exhibited similar DNA length increases in the presence of BLM, suggesting that the fluorescent dye does not affect BLM's unwinding activities (*Figure 1—figure supplement 2B*).

As nicks can be accidently generated along the long λ phage DNA template used in these assays, the observed dsDNA unwinding by BLM could be initiated either from a nick or from an intact duplex region melted internally, leading to distinct DNA intermediates during unwinding (*Rad et al., 2015*). To differentiate between these two possibilities, we analyzed the fluorescence signal of each individual unwinding event. In all 49 examined events from 33 traces, the unwinding progressed in one direction only, as reflected by the progression of the dark region of the fluorescence signal as well as the biased disappearance of its adjacent dsDNA (*Figure 1D*). This finding is not consistent with the possibility that the observed dsDNA unwinding was initiated by melting dsDNA internally where bidirectional progression of the unwinding fork movement was expected. Thus, we attributed these events to dsDNA unwinding initiated from a nick and the appearance and subsequent growth of the dark regions along the DNA track represented ssDNA. The rate of each unwinding fork movement was determined from the increase in ssDNA length revealed by the fluorescence signal and was converted to unwound base pair per second based on the elasticity of ssDNA (*Figure 1—figure supplement 3*). This yielded an unwinding rate of 60 ± 9 bp/s (mean ± SD) under 30 pN (*Figure 1E* and *Figure 1—source data 1*).

In conclusion, the BLM helicase at a high concentration is indeed able to perform unidirectional unwinding from a nick, but this requires exerting an external destabilizing force (tens of pN) on the DNA template.

## BLM translocates on intact ssDNA during unidirectional unwinding

Our data suggest that the BLM helicase unwinds dsDNA unidirectionally from a nick. This generates one intact strand under tension and one relaxed coiled strand in our experimental configuration. Since BLM unwinds by translocating on ssDNA in a 3' to 5' manner, we next sought to determine on which strand BLM prefers to translocate when initiating unwinding from a nick. This strand preference directly determines BLM's unwinding polarity from a single nick (*Figure 2A*). Thus, we designed a 6.4 kbp dsDNA harboring a single nick at 2.2 kbp and 4.2 kbp away from the 5' and the 3' ends, respectively (*Figure 2B* and *Figure 2—figure supplement 1*). BLM unwinds a 4.2 kbp dsDNA region if it translocates on the tensioned strand during unwinding or a 2.2 kbp dsDNA region if it translocates on the relaxed strand. To ensure the occurrence of DNA unwinding, we conducted the unwinding assay with this nicked DNA template under 30 pN. We found that 25 traces out of 29 examined DNA molecules showed continuous DNA unwinding from the nick. The 4.2 kbp dsDNA segments in these 25 traces were always unwound while the 2.2 kbp dsDNA segments remained intact, independent of the orientation of the suspended nicked DNA (*Figure 2B*). These findings suggest that BLM exclusively employed the intact ssDNA under tension as the track strand to unwind dsDNA. In addition, the inability of BLM to unwind the 2.2 kbp dsDNA segment reinforced our conclusion that BLM alone unwinds unidirectionally from a nick.

## RPA stimulates DNA unwinding by limited BLMs at low forces

Since RPA is a major protein partner that physically and functionally interacts with BLM, we next aimed to address how the human RPA (hRPA, hereafter referred to as RPA) regulates dsDNA unwinding by BLM. We first examined whether the presence of RPA could enhance BLM unwinding activity by reducing the required destabilizing force on the DNA template. To do so, we monitored helicase unwinding on the λ phage DNA in the presence of BLM and RPA while discretely increasing the external force (5, 10, 20 and 30 pN) until DNA unwinding was detected. Control experiments confirmed that RPA alone was incapable of unwinding dsDNA under these external forces (*Figure 3—figure supplement 1*). In the presence of 200 nM BLM and RPA at various concentrations,

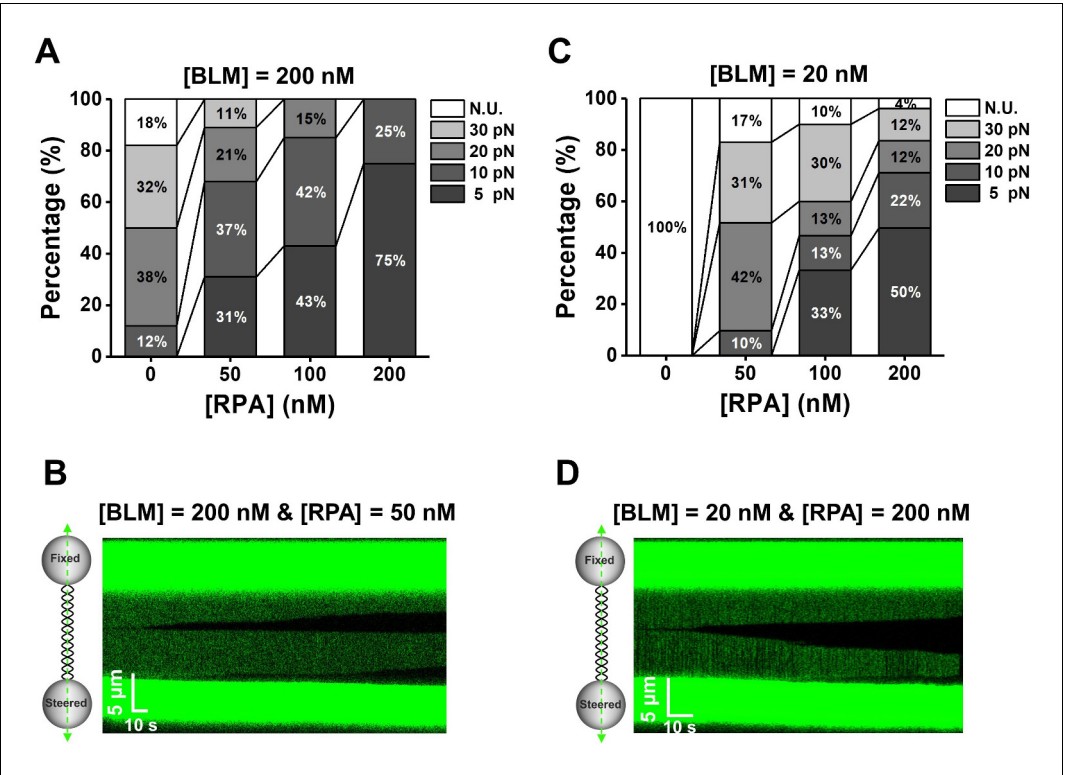

**Figure 3.** BLM unwinds dsDNA in the presence of RPA. (**A, C**) Distributions of the required forces to monitor DNA unwinding in the presence of 200 nM (**A**) or 20 nM (**C**) BLM and RPA with indicated concentrations. N.U. represents no unwinding. (**B, D**) Kymographs of tethered λ phage DNA showing DNA unwinding in the presence of 50 nM RPA and 200 nM BLM (**B**) or 200 nM RPA and 20 nM (**D**) BLM under 5 pN.

The online version of this article includes the following source data and figure supplement(s) for figure 3:

**Source data 1.** The required forces for BLM/RPA-mediated DNA unwinding.
**Figure supplement 1.** RPA alone was incapable of unwinding DNA.
**Figure supplement 2.** BLM and RPA unwind nicked DNA in the absence of external force.

DNA unwinding events initiated from the middle of the DNA template and the blunt-ended termini were observed (*Figure 3A*; *Figure 3B*). With the increase of RPA concentration, the required force to monitor DNA unwinding indeed significantly reduced (*Figure 3A* and *Figure 3—source data 1*). Moreover, we demonstrated that BLM/RPA-mediated DNA unwinding can even take place in the absence of any externally applied force (*Figure 3—figure supplement 2*).

Next, we asked whether a high abundance of BLMs are necessary for dsDNA unwinding when RPA is present. We assayed DNA unwinding of BLM at a concentration of 20 nM with various concentrations of RPA. In contrast to the observations with 20 nM BLM alone where no DNA unwinding was monitored from 10 to 40 pN at all (*Figures 1B* and *3C*), the presence of RPA can indeed stimulate BLM-mediated DNA unwinding in these conditions and the required force was consistently decreased with the increase of RPA concentration (*Figure 3C* and *Figure 3—source data 1*). It is noteworthy that with the presence of 200 nM RPA and 20 nM BLM, half of examined traces exhibited continuous DNA unwinding at a low force of 5 pN (*Figure 3C and D*).

Collectively, we conclude that RPA stimulates DNA unwinding of limited BLMs by lowering the required force on the DNA template.

## RPA activates BLM's bidirectional unwinding from a nick

We next sought to examine individual unwinding events in the presence of 50 nM RPA and 200 nM BLM and compare to those with 200 nM BLM alone. On average, there were 3.4 independent unwinding events monitored on a single DNA molecule, slightly higher than that without RPA (*Figure 1—figure supplement 1D*). Strikingly, careful analyses of fluorescence images on the λ phage

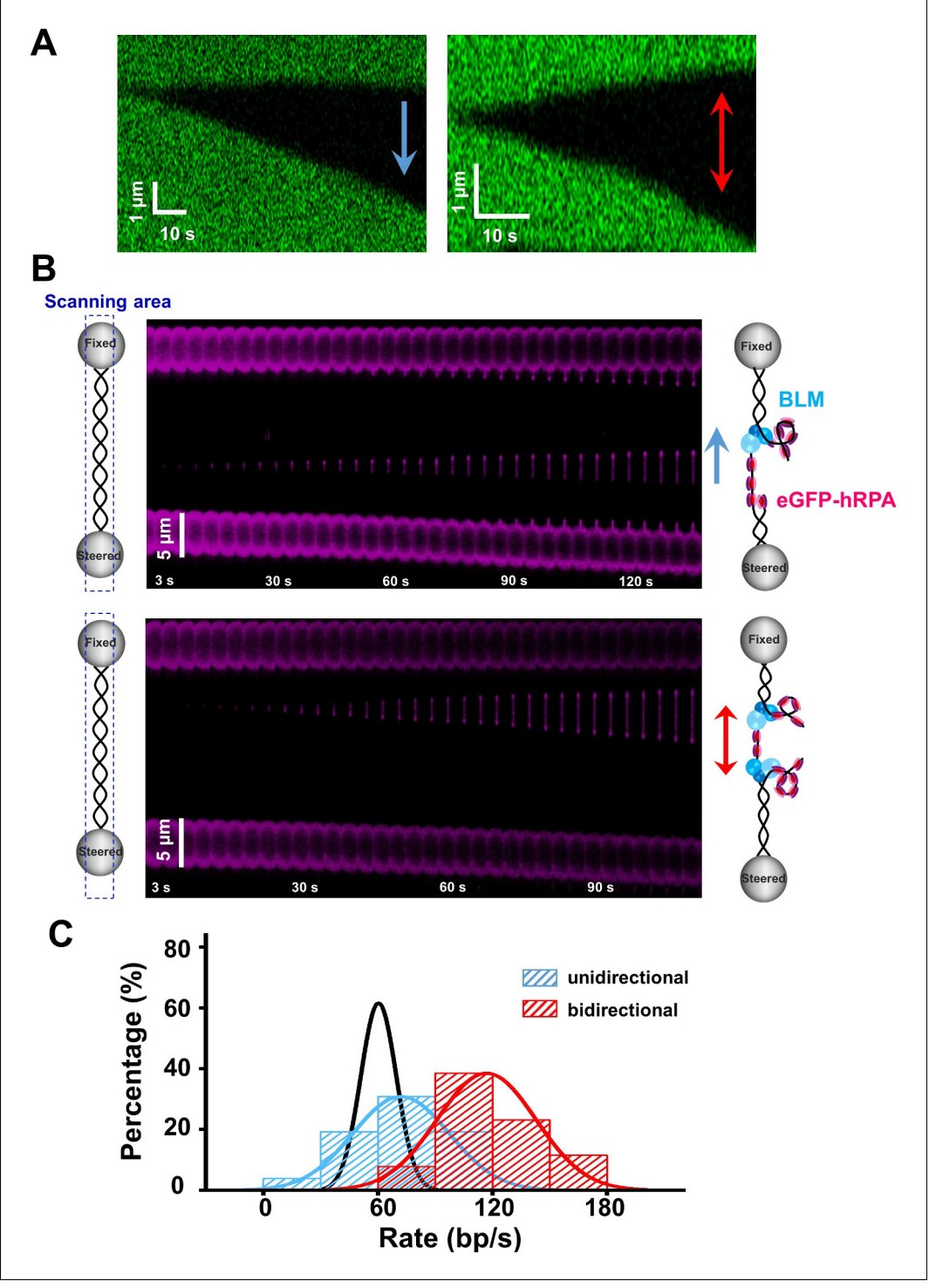

**Figure 4.** BLM's RPA-activated bidirectional unwinding from a nick. (**A**) Kymographs of individual unwinding events of a tethered λ phage DNA in the presence of 200 nM BLM and 50 nM RPA under 30 pN showing unidirectional and bidirectional DNA unwinding. The arrows indicate unwinding directions. (**B**) Representative kymographs of unidirectional (up) and bidirectional (down) DNA unwinding in the presence of 200 nM WT BLM and 50 nM eGFP-RPA. To illustrate the coiled ssDNAs, confocal laser scanned a rectangle area instead of only the DNA template track. The arrows indicate the directions of the unwinding fork movement. (**C**) Distributions of the uni- and bi- directional dsDNA unwinding rates in the presence of 200 nM BLM and 50 nM RPA under 30 pN. p-value<0.01 calculated by Student's *t* test. These rate distributions were compared with that in the presence of 200 nM BLM alone (black).

*Figure 4 continued on next page*

*Figure 4 continued*

The online version of this article includes the following source data and figure supplement(s) for figure 4:

**Source data 1.** The uni- and bi- directional dsDNA unwinding rates.
**Figure supplement 1.** RPA supports BLM's bidirectional unwinding on the 6.4 kbp nicked DNA template.
**Figure supplement 2.** RPA supports BLM's bidirectional unwinding under low forces.
**Figure supplement 3.** hBLM unwinds dsDNA in the presence and absence of hRPA.

DNA under a force of 30 pN revealed that, in addition to the expected unidirectional unwinding events, 53% of them (n = 40) showed bidirectional progression of dsDNA unwinding (*Figure 4A*). In line with these observations, 26 out of 27 traces from the experiments with the 6.4 kbp nicked DNA showed DNA unwinding in the presence of BLM and RPA and 35% of them also exhibited BLM's bidirectional unwinding initiated from the single nick on the template (*Figure 4—figure supplement 1*).

This RPA-activated bidirectional unwinding by BLM could be initiated from either internally melted dsDNA or a nick. To determine that, we labeled the RPA with enhanced green fluorescent protein (eGFP) and repeated the unwinding experiments. This eGFP-RPA allowed for the direct visualization of the unwinding intermediates of ssDNA instead of dsDNA (*King et al., 2013*). If dsDNA unwinding is initiated from melted regions of dsDNA, two continuous ssDNA strands under tension would be expected, and no frayed relaxed ssDNA exists. However, in most examined unwinding events (90%, n = 95), we observed at least one bright spot locating at the end of the newly generated tensioned ssDNA under 30 pN (*Figure 4B*). These spots increased in intensity as the unwinding fork progressed, and are thus attributed to unwound coiled ssDNA coated by RPA. These observations support the interpretation that the RPA-stimulated DNA unwinding by BLM is initiated from a nick instead of internally melted dsDNA regions. Moreover, 23 unwinding events were marked with two spots on both ends of an ssDNA, progressing in opposite directions (*Figure 4B*). These observations can be interpreted as BLM helicases unwind two opposite DNA forks by translocating on tensioned ssDNA in one direction and on relaxed ssDNA in the other, further strengthening our conclusion that that RPA activates BLM's bidirectional unwinding. Intriguingly, both Sytox and eGFP signals indicated that the bidirectional unwinding only initiates from a nick instead of starting an opposite unwinding direction in the middle of an ongoing unidirectional unwinding. The average bidirectional unwinding rate (117 ± 25 bp/s, mean ± SD) was 1.6-fold higher than that of the unidirectional unwinding (72 ± 24 bp/s, mean ± SD) which is slightly faster than that with BLM alone (60 ± 9 bp/s, mean ± SD) (*Figure 4C* and *Figure 4—source data 1*). These findings indicate that the unwinding rates for both directions might be a little different. As the opposite DNA unwinding forks differ in that BLMs translocate on relaxed and tensioned ssDNAs, we reasoned that the tension on the ssDNA might modulate BLM unwinding activity a bit. Notably, bidirectional DNA unwinding events were also observable under low and zero external forces (*Figure 3—figure supplement 2* and *Figure 4—figure supplement 2*).

To rule out that the observed bidirectional DNA unwinding was due to the heterologous combination of gBLM and hRPA, we also conducted the DNA unwinding experiments by using hBLM. When testing the homologous pair of hBLM and hRPA, results similar to the heterologous combination were obtained: hBLM alone initiates unidirectional unwinding from a nick and hRPA activates its bidirectional unwinding (*Figure 4—figure supplement 3*).

## RPA can also activate core-BLM's bidirectional unwinding

To examine whether BLM's RPA-activated bidirectional unwinding is due to the interactions between them, we utilized a core fragment of *Gallus gallus* BLM that lacks major interaction domains with RPA (hereafter referred to as core-BLM) (*Figure 5A*; *Doherty et al., 2005*; *Kang et al., 2018*). The WT BLM helicase was replaced with core-BLM mutant in the helicase unwinding assay with the λ DNA. Control experiments verified that core-BLM alone also unidirectionally unwinds dsDNA from a nick with a relatively slower rate compared with WT BLM (*Figure 5—figure supplement 1A–C*). When RPA was present, both the unidirectional (58%) and bidirectional (42%) unwinding events (n = 45) by this mutant were recorded (*Figure 5B*). These observations were further substantiated by experiments using eGFP-RPA (*Figure 5C*) and the 6.4 kbp DNA template (32%, n = 37) (*Figure 5—*

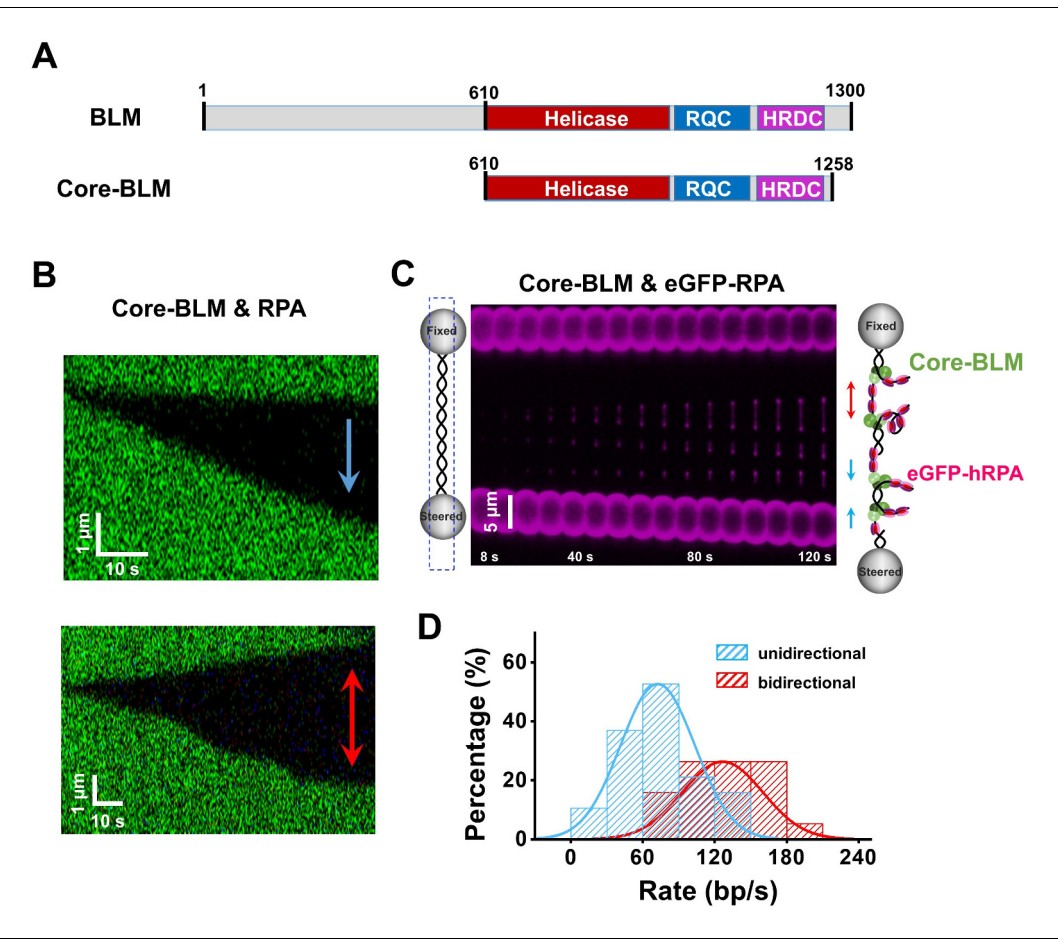

**Figure 5.** RPA supports core-BLM's bidirectional unwinding from a nick. (**A**) Schematic representations of WT BLM and core-BLM. Helicase, RQC and HRDC domains are shown in different colors. (**B**) Kymograph of an individual unwinding event in the presence of 20 nM core-BLM and 50 nM RPA under 35 pN showing unidirectional (up) and bidirectional (down) DNA unwinding. The arrows indicate the directions of the unwinding fork movement. (**C**) A representative kymograph of a single DNA tether unwound in the presence of 20 nM core-BLM and 50 nM eGFP-RPA. To demonstrate the coiled ssDNAs, confocal laser scanned a rectangle area instead of the DNA template track only. (**D**) Distributions of the uni- and bi- directional dsDNA unwinding rates in the presence of 20 nM BLM and 50 nM RPA under 35 pN. p-value<0.01 calculated by Student's *t* test.

The online version of this article includes the following source data and figure supplement(s) for figure 5:

**Source data 1.** The uni- and bi- directional dsDNA unwinding rates.

**Figure supplement 1.** Core-BLM unwinds dsDNA in the presence and absence of RPA.

---

*figure supplement 1D*). The rate of the bidirectional unwinding is 1.7-fold higher than that of the unidirectional unwinding, in agreement with the results with WT BLM (*Figures 4C* and *5D* and *Figure 5—source data 1*). These findings resemble the observations with WT BLM, reflecting that BLM's RPA-activated bidirectional unwinding possibly does not rely on the interactions between them.

## Free RPAs in solution permits BLM's continuous unwinding

Having demonstrated that RPA activates BLM's bidirectional unwinding, we next aimed to gain mechanistic insights into this activation. A few potential mechanisms have been proposed to explain how RPA stimulates helicase unwinding of long duplex dsDNA *Awate and Brosh (2017)*. RPA may enhance BLM-catalyzed DNA unwinding by coating the unwound ssDNAs and relieving the pressure from their reannealing to prolong BLM tethering at the ssDNA/dsDNA junction during unwinding of relatively long duplexes. Alternatively, RPA coated onto ssDNA at the fork junction may help recruit

additional BLM helicases from solution to aid in duplex unwinding. To examine these possibilities, we conducted a BLM unwinding experiment in a microfluidic system which allows for the rapid switching of experimental conditions (*Figure 6—figure supplement 1*). In one channel, we first started the dsDNA unwinding by using 20 nM BLM and 50 nM RPA at 30 pN to ensure the initiation of dsDNA unwinding from a nick. Following that, we depleted the free proteins in solution by rapidly transporting the DNA tether to an ATP-containing reaction buffer channel. We found that, in the buffer channel, the DNA length slowly decreased and the dark region along the DNA template gradually disappeared, indicating the reannealing of the unwound ssDNA (n = 30, *Figure 6A*). This transition from DNA unwinding to rewinding is a result of the dissociation of ssDNA coated RPA and is possibly due to BLM strand switching and/or dissociation. This observation suggests that either free RPAs or BLMs are indispensable for the processive unwinding. To further determine that, we transported BLM/RPA-initiated DNA unwinding template to a channel containing either BLM or RPA. In these experiments, we observed that up to 75% of traces (n = 53) in the following RPA channel continued DNA unwinding, and yet only 16% of traces showed continuous unwinding in the BLM channel (n = 38) (*Figure 6B and C*). We thus conclude that free RPAs in solution are indispensable for promoting continuous dsDNA unwinding by BLM.

Finding that only free RPA is necessary for BLM-mediated dsDNA unwinding once initiated raises a possibility that BLM might be needed to recognize a nick and bind to it. To test this hypothesis, we started the experiment by first placing the dsDNA tether in a buffer containing 20 nM BLM only where no dsDNA unwinding occurred (*Figure 6D*). Sequentially, the DNA tether was rapidly switched to a channel containing 50 nM RPA. In this channel, we immediately observed continued unwinding from both DNA length and fluorescence signals in 21 out of 32 traces, resembling the unwinding traces observed when both proteins were present at the same time (*Figure 6D*). Depleting ATP in the RPA channel abolished the unwinding, confirming that these observations relied on BLM unwinding activity (*Figure 6—figure supplement 1B*). In addition, 17% of the unwinding events (n = 12) were also observed to progress bidirectionally in this experiment, further supporting that this observed unwinding was due to the presence of both BLM and RPA. Since the BLM helicase has poor binding ability to intact dsDNA (*Shi et al., 2017b*), the immediate unwinding observed in the RPA channel suggest that limited BLMs recognize and bind to nicks along the DNA template in the BLM channel, and they are stimulated by RPA to unwind a substantial distance without dissociation. To further examine whether DNA-bound RPA could recruit BLM to the DNA template, we transported RPA-coated DNA template to the BLM and RPA channel. Only 17% of unwinding events (n = 33) showed continued unwinding in this channel (*Figure 6—figure supplement 2*), in disagreement with the recruitment model.

In summary, we conclude that once BLM recognizes a nick and initiates unwinding with RPA, only free RPAs are indispensable for its processive unwinding. These findings favor the model that ssDNA generated by BLM was immediately coated by free RPA, preventing the separated ssDNAs from reannealing.

## Discussion

There are several lines of evidence supporting that unwinding activities of the RecQ family helicases can be regulated by the presence of SSB proteins (*Awate and Brosh, 2017*; *Bagchi et al., 2018*; *Cui et al., 2004*; *Mills et al., 2017*; *Shen et al., 1998*; *Xue et al., 2019a*). Herein, our experimental approach combining optical tweezers with fluorescence microscopy has shed new light on the effect of RPA on BLM's ability to catalyze dsDNA unwinding. This combined technique allows for the direct monitoring of individual unwinding events by BLM along tensioned dsDNA in real time, leading to the discovery of a distinct unwinding mode. In this mode, RPA activates BLM's bidirectional unwinding from a nick where BLMs can translocate on both relaxed and tensioned ssDNAs during unwinding. This is the first observation, to our knowledge, of helicase bidirectional unwinding from a nick. Previous studies of replicative and DNA repair helicases have reported bidirectional unwinding attributable to helicase melting dsDNA internally, which differs from our observed results (*Rad et al., 2015*; *Yardimci et al., 2012*).

In the absence of RPA, BLM alone can only initiate unidirectional dsDNA unwinding from a nick where it translocates on the ssDNA with the assistance of an external destabilizing force on DNA (*Figure 1D*, *Figure 4—figure supplement 3*, and *Figure 5—figure supplement 1*). Since the

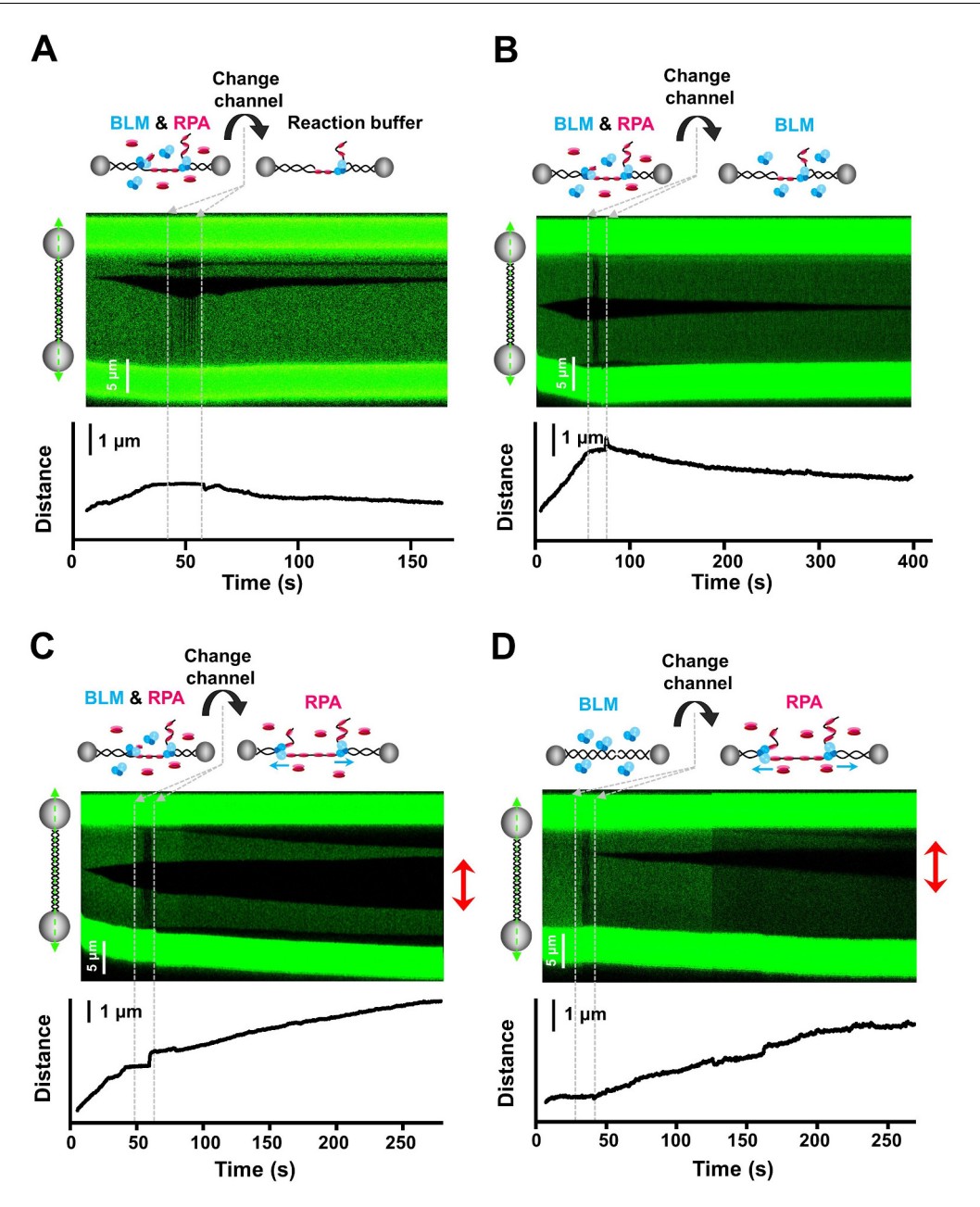

**Figure 6.** Free RPAs are indispensable for stimulating BLM-mediated DNA unwinding. (**A–C**) Kymographs and DNA length vs. time traces of DNA tethers showing dsDNA unwinding initiating in a channel containing 20 nM BLM and 50 nM RPA under 30 pN, followed by quickly transporting to a channel containing ATP reaction buffer (**A**), 20 nM BLM (**B**) or 50 nM RPA (**C**) only. Red arrows indicate bidirectional DNA unwinding. (**D**) Kymograph and DNA length vs. time of a DNA tether showing the DNA template incubated in a channel containing 20 nM BLM under 30 pN, followed by transporting to a channel containing 50 nM RPA. Red arrows indicate bidirectional DNA unwinding.

The online version of this article includes the following figure supplement(s) for figure 6:

**Figure supplement 1.** The micro-flow cell assay confirmed the observe DNA unwinding requires BLM's unwinding activity.

**Figure supplement 2.** RPA cannot recruit BLM to the DNA unwinding fork.

physical barriers presented by dsDNA on both directions from a nick are similar, the inability of BLM

to unwind in the opposite direction is mostly likely due to the lack of tension on the relaxed ssDNA which might possibly promote the unwinding activity of BLM. This unidirectional unwinding on nicked dsDNA by BLM alone occurred with the high protein concentrations. The core-BLM lacking the oligomerization domain also exhibited unidirectional unwinding from a nick in the absence of RPA (*Figure 5—figure supplement 1*; *Karow et al., 1999*; *Shi et al., 2017a*), leading us to exclude the possibility that processive unwinding could depend on oligomerization of the protein. Instead, proteins at high concentrations might ensure sufficient binding of the enzyme to the ssDNA generated during an unwinding event, allowing for successive rounds of unwinding and preventing the unwound strands against reannealing.

A distinct feature of the bidirectional dsDNA unwinding by BLM and RPA is that the unwinding directionality is determined initially at a nick (*Figure 4*). The sequential protein incubation assays demonstrated that, once preloaded with BLM, only free RPAs are indispensable for activating the bidirectional dsDNA unwinding, suggesting that only limited BLMs are required to first recognize and load on the nicked dsDNA (*Figure 6C and D*). Thus, we reasoned that the bidirectional unwinding necessitates the preloading of more than one helicase at the nick and subsequent binding of RPA on the newly generated ssDNA prolongs the binding and unwinding of BLM at the fork. The unidirectional unwinding probably results from only one BLM helicase preloading at the nick and the inability of BLM to associate with RPA-coated ssDNA (*Xue et al., 2019a*). This speculation could also explain a recent single-molecule observation that only unidirectional DNA unwinding by BLM and RPA was detected wherein low concentrations of BLM were used (*Xue et al., 2019a*).

Our study also provided insights into the mechanisms by which BLM and RPA cooperate to facilitate end resection in HR. Recent studies support a model that the long-range end resection in HR initiates from a nick generated by the MRN/X complex once the DSB ends are coated by DNA adducts (*Cannavo and Cejka, 2014*). This demands that the helicase and the nuclease unwind and resect bidirectionally from a nick. It is highly likely that the abilities of BLM to recognize and preload on the nick would help identify the locations of DSB and possibly play a role in recruiting other required proteins for the resection. Although it needs to be further validated, the inherent ability of BLM and RPA to unwind bidirectionally from a nick would facilitate the resection by DNA adduct coated DSB ends which provides ssDNA for the DNA2/EXO1 nucleases to degrade from 5' to 3' and the MRN complex to degrade from 3' to 5' (*Symington, 2016*). Interestingly, inhibition of MRN exonuclease activity only confers a relatively lower resection defect compared with the inhibition of its endonuclease activity, suggesting that other proteins might also participate in the initial short ssDNA generation (*Shibata et al., 2014*). In support of this notion, the complete unwinding of the short dsDNA between the nick and the DSB end may serve as an additional pathway in generating the short stretches of ssDNA.

The finding that the SSB proteins promote helicase-catalyzed dsDNA unwinding from a nick might be conserved among the RecQ-like helicases. For example, the RecQ helicase from *E. coli* is also able to promote dsDNA unwinding in both directions in the presence of *E. coli* SSB (*Rad et al., 2015*). However, its bidirectional unwinding was initiated by melting dsDNA internally instead of a nick, which resembles replicative helicases in DNA replication initiation (*Mott and Berger, 2007*). In fact, the unwinding by RecQ progresses unidirectionally if initiated from a nick. In addition, the Sgs1 helicase, the S*accharomyces cerevisiae* homologue of BLM, was also found to unwind dsDNA from blunt-ended termini or a nick with RPA (*Wang et al., 2018*; *Xue et al., 2019b*), but its unwinding directionality when initiated from a nick has not been determined. A previous single-molecule FRET study revealed that BLM could not unwind more than 34 bp even in the presence of RPA (*Yodh et al., 2009*). This difference may be due to the fact that the action of applied force on the DNA template can lower the energy required to melt the DNA and prevent reannealing of the separated strands, thus stimulating BLM's unwinding activity. Given the fact that hRPA can activate hBLM's, gBLM's and core-gBLM's bidirectional unwinding form a nick (*Figures 4* and *5* and *Figure 4—figure supplement 3*), this activation might be a conserved feature for the SSB proteins and the RecQ helicases and does not require a homologous system or physical interactions between them.

# Materials and methods

**Key resources table**

| Reagent type (species) or resource | Designation | Source or reference | Identifiers | Additional information |
|---|---|---|---|---|
| Gene (*Gallus gallus*) | *gBLM* | NCBI | RRID:SCR:006472 | NP_001007088.2 |
| Gene (*Homo sapiens*) | *hBLM* | NCBI | RRID:SCR:006472 | NP_000048.1 |
| Strain, strain background (*Escherichia coli*) | 2566 | This paper | | Competent cell |
| Strain, strain background (*Escherichia coli*) | BL21(DE3) | This Paper | | Competent cell |
| Recombinant DNA reagent | Lambda DNA | Thermo Fisher Scientific | SD0021 | |
| Recombinant DNA reagent | pBR322 (plasmid) | Takara | RRID: Addgene_10877 | |
| Recombinant DNA reagent | pTWIN1 (plasmid) | New England BioLabs | N6951S | Expression of *core BLM* in *E. coli* |
| Recombinant DNA reagent | pET21a-sumo (plasmid) | This paper | | Expression of BLM in *E. coli* |
| Recombinant DNA reagent | p11d-tRPA (plasmid) | Vector Builder | Addgene:102613 | Expression of *human RPA* in *E. coli* |
| Peptide, recombinant protein | T4 DNA ligase | New England BioLabs | M0202L | |
| Peptide, recombinant protein | Klenow Fragment, exo- | Thermo Fisher Scientific | EP0421 | |
| Peptide, recombinant protein | Nde I | New England BioLabs | R0111S | |
| Peptide, recombinant protein | Sal I | New England BioLabs | R3138S | |
| Peptide, recombinant protein | BstX I | New England BioLabs | R0113L | |
| Chemical compound, drug | Biotin-dATP | Invitrogen | Invitrogen: 19524016 | |
| Chemical compound, drug | Biotin-dCTP | Invitrogen | Invitrogen: 19518018 | |
| Chemical compound, drug | IPTG | Thermo Fisher Scientific | 15529019 | |
| Commercial assay, kit | Streptavidin Coated Polystyrene Particles (1.76 µm) | Spherotech | AG02 | |
| Commercial assay, kit | Streptavidin Coated Polystyrene Particles (4.42 µm) | Spherotech | AL01 | |
| Commercial assay, kit | DNA Purification | Sangon Biotech | B518141 | |
| Commercial assay, kit | SP Sepharose Fast Flow | GE Healthcare | 17072904 | |
| Commercial assay, kit | Q Sepharose Fast Flow | GE Healthcare | 17051004 | |

*Continued on next page*

*Continued*

| Reagent type (species) or resource | Designation | Source or reference | Identifiers | Additional information |
|---|---|---|---|---|
| Commercial assay, kit | Sytox orange Nucleic Acid Stain | Thermo Fisher Scientific | S11368 | 5 mM |
| Software, algorithm | MATLAB, Data analysis | MathWorks | RRID: SCR:001622 | |

## Preparation of DNA templates

The λ phage DNA template was constructed as described elsewhere (*Gross et al., 2010*). Briefly, biotinylated λ phage DNA was made through 3'-end labeling by fill-in of 5'-overhangs with an exo-Klenow Fragment. The reaction was conducted by incubating 330 nM λ phage DNA, 600 μM dGTP/dATP/dTTP, 400 μM biotin-14-dCTP, and 5U Klenow in 1X Klenow reaction buffer at 37°C for 1 hr. The mixture was purified by Column DNA Purification Kit.

The 6.4 kbp DNA template containing a single nick consisted of two DNA segments connected by an adaptor (*Figure 2—figure supplement 1*). The two 2.2 kbp and 4.2 kbp DNA segments were PCR amplified from the plasmid pBR322 using a biotin-labeled primer. The resulting DNA fragments were digested with BstXI to create an overhang. The adaptor was produced by annealing three oligonucleotides where a nick was automatically generated because the 5' phosphate of one oligonucleotide was absent (*Figure 2—figure supplement 1*). The final product was produced by ligating the two DNA segments with the adaptor at 1:1:1 ratio using T4 ligase.

## Protein purification

### BLM

The wild type *Gallus gallus* bloom syndrome protein (gBLM, referred to as WT BLM) and its helicase core mutant (BLM[610-1258], referred to as core-BLM) were used in our experiments. Both WT BLM and core-BLM were expressed and purified as previously described (*Shi et al., 2017a*). In brief, the BLM gene was amplified and constructed into the pET21a-sumo vector. The N-terminal-domain-truncated core-BLM was constructed into pTWIN1 by using Nde I and Sal I restriction sites, and transformed into BL21 (DE3). Expression was induced in the T7 expressing *E. coli* strain 2566 by 0.3 mM isopropyl 1-thio-D-galactopyranoside at 18°C for 16 hr. Protein was homogenously purified sequentially by affinity chromatography with a complete His tag purification resin column and ion exchange chromatography on SP Sepharose Fast Flow and Q-Sepharose Fast Flow respectively. hBLM was purified as described previously (*Karow et al., 1997*).

### hRPA

The human RPA plasmid was a gift from Dr. Marc Wold. The *E. coli* strain BL21 (DE3) was transformed with the plasmid p11d-tRPA, permitting the co-expression of RPA70, RPA32, and RPA14. RPA was then purified over Affi-Gel Blue, Hydroxyapatite (Biorad), and Q-Sepharose chromatography columns as described previously (*Henricksen et al., 1994*). The purified protein was eluted in phosphate buffer containing 300 mM KCl (pH 7.5). To obtain fluorescent human RPA, a DNA fragment encoding a variant of the enhanced GFP (eGFP) with a polyhistidine tag was inserted in frame at the 3' end of the cDNA encoding the large subunit of RPA in the expression plasmid p11d-tRNA. eGFP-RPA purification was performed as previously described (*van Mameren et al., 2009*).

## Single-molecule experiments

Single-molecule experiments were performed at 25°C on an instrument combining three-color confocal fluorescence microscopy with dual optical traps (LUMICKS, C-trap). In brief, a 1,064 nm fiber laser and a water-immersion objective were used to create two orthogonally polarized optical traps. The trap separation was controlled using a piezo mirror for beam-steering one trap. Force measurements were performed by back-focal plane interferometry of the condenser top lens using a position-sensitive detector. A computer-controlled stage enabled rapid movement of the optical traps within a multiple-channel flow cell (*Figure 6—figure supplement 1*). This flow cell allowed for the rapid in situ construction and characterization of dumbbell constructs, and facilitated the swift and complete transfer of the tethered DNA between different flow channels.

As described previously (*Gross et al., 2010*), DNA molecules were captured between two streptavidin-coated polystyrene beads (1.76 µm for 6.4 kbp DNA and 4.5 µm for λ phage DNA) using the multichannel laminar flow cell and tensioned by increasing the distance between the optical traps (*Figure 6—figure supplement 1*). A single DNA was verified by its inherent mechanical force-extension curve. The trap was then moved to protein channels as described for each assay. Unless stated otherwise, all experiments were carried out in a reaction buffer containing 25 mM Tris-HCl pH 7.5, 100 mM NaCl, 1 mM $MgCl_2$, 0.1 mg/ml BSA, 2 mM ATP, 3 mM DTT and 50 nM Sytox Orange.

Fluorescence microscopy was achieved by imaging the stained DNA on an EMCCD camera. Here, a 488 nm excitation laser and a 532 nm excitation laser were used for imaging eGFP-hRPA and Sytox Orange respectively. Kymographs were generated via a confocal line scan through the center of the two beads.

## Data acquisition and analysis

Single-molecule force and fluorescence data were analyzed using custom software provided by LUMICKS (available at http://www.nat.vu.nl/~iheller/download.html). Force and DNA extension data were taken at 50 kHz and filtered to 30 Hz. The kymograph was used to track the edge of the unwinding forks. The edges were marked manually by examining each pixel image and comparing to a set threshold value. Elasticity parameters of ssDNA were obtained from the DNA force-extension measurements for data conversion. The force-extension relation of ssDNA in our experimental condition was described as an extensible freely jointed-chain model (*Figure 1—figure supplement 3A*; *Smith et al., 1996*). The unwinding rate was obtained from fitting to the linear region of the increase of the unwinding fork versus time (*Figure 1—figure supplement 3B*). The unwinding rates were reported as the mean ± S.D. from the indicated number of events.

## Acknowledgements

This research was supported by the National Key R and D Program of China (2016YFA0500902 and 2017YFA0106700), the Natural Science Foundation of Shanghai (19ZR1434100), the French National Cancer Institute (PLBIO2017-167), and the French National League Against Cancer (EL2028.LNCC/MaM).

## Additional information

### Funding

| Funder | Grant reference number | Author |
|---|---|---|
| Ministry of Science and Technology of the People's Republic of China | 2016YFA0500902 | Bo Sun |
| Ministry of Science and Technology of the People's Republic of China | 2017YFA0106700 | Bo Sun |
| Shanghai Natural Science Foundation | 19ZR1434100 | Bo Sun |
| Institut National Du Cancer | PLBIO2017-167 | Mauro Modesti |
| The French National League Against Cancer | EL2028.LNCC/MaM | Mauro Modesti |

The funders had no role in study design, data collection and interpretation, or the decision to submit the work for publication.

### Author contributions

Zhenheng Qin, Lulu Bi, Data curation, Software, Formal analysis, Validation, Visualization, Methodology, Writing - original draft, Writing - review and editing; Xi-Miao Hou, Mauro Modesti, Resources, Writing - review and editing; Siqi Zhang, Resources, Methodology; Xia Zhang, Resources, Data curation, Software, Methodology; Ying Lu, Ming Li, Resources, Writing - original draft; Xu-Guang Xi,

Conceptualization, Resources, Supervision, Project administration, Writing - review and editing; Bo Sun, Conceptualization, Resources, Software, Formal analysis, Supervision, Funding acquisition, Visualization, Methodology, Writing - original draft, Project administration, Writing - review and editing

### Author ORCIDs
Mauro Modesti http://orcid.org/0000-0002-4964-331X
Bo Sun https://orcid.org/0000-0002-4590-7795

### Decision letter and Author response
Decision letter https://doi.org/10.7554/eLife.54098.sa1
Author response https://doi.org/10.7554/eLife.54098.sa2

## Additional files
### Supplementary files
• Transparent reporting form

### Data availability
All data generated or analysed during this study are included in the manuscript and supporting files. Source data files have been provided for Figures 1, 3, 4 and 5.

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
