## [Decision Letter]

**Acceptance summary:**

The cooperation between RecQ family helicases and ssDNA binding proteins, such as RPA, are important for many aspects of DNA repair. By combining force-extension and fluorescence measurements, the authors have built a compelling case that in the presence of RPA BLM helicase can perform a "bidirectional" unwinding from a nick in the dsDNA unwinding the duplex upstream and downstream of the nick. This observation illustrates the activity of the BLM helicase in long-range dsDNA end resection during initiation of homologous recombination.

**Decision letter after peer review:**

Thank you for submitting your article "RPA Activates BLM's Bidirectional DNA Unwinding from a Nick" for consideration by *eLife*. Your article has been reviewed by three peer reviewers, and the evaluation has been overseen by a Reviewing Editor and John Kuriyan as the Senior Editor. The following individual involved in review of your submission has agreed to reveal their identity: Mihály Kovács (Reviewer #2).

The reviewers have discussed the reviews with one another and the Reviewing Editor has drafted this decision to help you prepare a revised submission.

Summary:

In this study the authors performed optical tweezers-based single-molecule experiments combined with fluorescence microscopy to demonstrate that Bloom's syndrome (BLM) helicase, in conjunction with Replication Protein A (RPA), is able to perform bidirectional DNA unwinding starting from a nick located on one of the DNA strands.

Essential revisions:

The three reviewers and the reviewing editor came to an agreement that you have built a compelling case that in the presence of RPA BLM helicase can perform a "bidirectional" unwinding from a nick in the dsDNA, i.e. it can move on both, the relaxed strand and on the strand under tension. Combination of the force-extension and fluorescence measurements allowed you to quantify the rate, the frequency and directionality of the unwinding events. An impressive set of experimental conditions has been sampled to identify the circumstances under which the gBLM activity can be observed on the nicked DNA.

The reviewers, however identified a number of issues with the paper that questions its suitability, at least in its current state.

Major issues that require new experiments:

1) The first significant problem lays in the use of heterologous BLM (from bird) and RPA (human). The observations are compared to what is known from the human system, but the gBLM itself does not seem to have a characteristic unwinding behavior of the human enzyme or other RecQ helicases, which is typically very dynamic with repetitive events and backtracking. Furthermore, the likely absence of the gBLM-hRPA interaction reduces the physiological significance of the authors' observations. While specific interactions between the two proteins may not be necessary for the observed reactions, the presence of such an interaction may alter the BLM behavior. The authors need to make a better case for the physiological relevance of their observations or to add an analysis of the cognate protein pair for at least the key experiments. The necessity of the experiments with human BLM is a major experiment that needs to be added to strengthen the conclusions of the manuscript. It has been requested by all three reviewers and the reviewing editor.

2) A complementary biochemical in vitro experiment is necessary to establish that the bidirectional unwinding can indeed take place in the absence of force. Extrapolation of the rate of initiation dependence on force to zero force may be another way to address this question.

Other important issues to address:

3) It is unclear how many nicks are present on each λ DNA molecule, which complicates the analysis of the initiation rate.

4) The authors should support their statements with statistical analyses (p values, errors, etc), especially considering a relatively small number of events in each class.

---

## [Author Response]

Essential revisions:[…] Major issues that require new experiments:1) The first significant problem lays in the use of heterologous BLM (from bird) and RPA (human). The observations are compared to what is known from the human system, but the gBLM itself does not seem to have a characteristic unwinding behavior of the human enzyme or other RecQ helicases, which is typically very dynamic with repetitive events and backtracking. Furthermore, the likely absence of the gBLM-hRPA interaction reduces the physiological significance of the authors' observations. While specific interactions between the two proteins may not be necessary for the observed reactions, the presence of such an interaction may alter the BLM behavior. The authors need to make a better case for the physiological relevance of their observations or to add an analysis of the cognate protein pair for at least the key experiments. The necessity of the experiments with human BLM is a major experiment that needs to be added to strengthen the conclusions of the manuscript. It has been requested by all three reviewers and the reviewing editor.

We greatly appreciate the comment and suggestion from the reviewers and the editor and would like to briefly comment on why we chose to use gBLM in our previous manuscript. First, gBLM is an ortholog of human BLM (hBLM) and shares high similarity with hBLM. gBLM has two similar RPA binding domains to the hBLM (Kang et al. 2018) and the sequence of the core gBLM has an 80% identity with the one from hBLM. Second, we have successfully over-expressed and purified gBLM with high purity (Shi et al., *2017*) and biochemical studies with gBLM also showed similar results with hBLM (Shi et al., 2017). Thus, we chose to mainly use gBLM in our previous study to investigate how hRPA stimulates its unwinding.

After reading the comment, we agree with the reviewers and the editor on that the experiments with hBLM are more physiologically relevant and necessary to further strengthen our conclusions. Following that, we have conducted an additional set of experiments. Frist, we performed a similar DNA unwinding assay with WT hBLM alone and observed only unidirectional unwinding events initiated from nicks. Second, we conducted experiments with hBLM and hRPA. In this assay, a similar bidirectional DNA unwinding was observable by using Sytox as a probe. Third, to further verify the bidirectional unwinding is initiated from a nick. We used eGFP-hRPA to indicate the unwinding intermediates of ssDNA in the presence of hBLM. Consistently, the observed bidirectional unwinding events were often accompanied with one or two bright spots at the end of the newly generated ssDNA, indicating that the DNA unwinding was also initiated from a nick in this experimental condition. This new set of experiments with hBLM reinforces our findings that RPA can activate BLM’s bidirectional unwinding from a nick and rules out the observed bidirectional DNA unwinding was an artificial result of the heterologous combination of gBLM and hRPA. We have added this set of data as Figure 4—figure supplement 3 and also incorporated this information in the revised manuscript by stating:

“To rule out that the observed bidirectional DNA unwinding was due to the heterologous combination of gBLM and hRPA, we also conducted the DNA unwinding experiments by using hBLM. When testing the homologous pair of hBLM and hRPA, results similar to the heterologous combination were obtained: hBLM alone initiates unidirectional unwinding from a nick and hRPA activates its bidirectional unwinding (Figure 4—figure supplement 3).”2) A complementary biochemical in vitro experiment is necessary to establish that the bidirectional unwinding can indeed take place in the absence of force. Extrapolation of the rate of initiation dependence on force to zero force may be another way to address this question.

We greatly appreciate the reviewers’ suggestion and would be more than happy to answer this question by conducting new experiments. We first tried to address whether BLM and RPA can unwind DNA in the absence of any externally applied force by employing an in vitro ensemble assay. However, as BLM and RPA can also initiate the DNA unwinding from the blunt-ended termini (Figure 3B) which raises difficulties in the experimental design and data analysis. Instead, we designed a single-molecule assay to answer this question. In this assay, we first incubated a relaxed DNA tether (under zero force) in a BLM and eGFP-hRPA channel for a short period time. Following that, we transported the DNA tether to a buffer channel wherein no unwinding occurred because of the absence of BLM, RPA and ATP. In this channel, we examined the fluorescence signal of eGFP-hRPA along the DNA tether under 10 pN. We found the existence of the unwound ssDNA on the tensioned DNA as well as one or two bright spots at the end of the newly generated ssDNA. This results strongly suggest that BLM and RPA can indeed unwind dsDNA from a nick in the absence of external force. We have added this set of data as Figure 3—figure supplement 2 and also incorporated this information in the revised manuscript by stating:

“Moreover, we demonstrated that BLM/RPA-mediated DNA unwinding can even take place in the absence of any externally applied force (Figure 3—figure supplement 2).” and “Notably, bidirectional DNA unwinding events were also observable under low and zero external forces (Figure 3—figure supplement 2 and Figure 4—figure supplement 2).”Other important issues to address:3) It is unclear how many nicks are present on each λ DNA molecule, which complicates the analysis of the initiation rate.

We apologize for not being explicit on describing our experiment. For the experiments with λ DNA, we can detect unwinding events along the template but not the number of nicks on it. In other words, the nicks that were not used to initiate dsDNA unwinding cannot be detected. Thus, it is impossible to analyze the initiation rate by using this DNA template. However, we can indeed calculate the initiation rate by the experiments with the 6.4 kbp DNA template which contains a single nick. In the presence of BLM alone, 25 traces in all 29 examined DNA tethers (86%) were found to be unwound from the single nick. In the presence of BLM and RPA, we found that nearly all examined DNA molecules (26 out of 27 examined traces) were unwound from the nick. This information was provided in the revised manuscript in the subsection “RPA activates BLM’s bidirectional unwinding from a nick”.

4) The authors should support their statements with statistical analyses (p values, errors, etc), especially considering a relatively small number of events in each class.

We thank the reviewer for the suggestion. We have incorporated them in the updated manuscript and provided source data as well. Please note that p-values have been provided in the revised figure legends.